# A Dual Forward–Backward Algorithm to Solve Convex Model Predictive Control for Obstacle Avoidance in a Logistics Scenario

**Daniele Ludovico** [1,*,†], **Paolo Guardiani** [1,†], **Alessandro Pistone** [1,†], **Lorenzo De Mari Casareto Dal Verme** [1,2], **Darwin G. Caldwell** [1] and **Carlo Canali** [1]

1   Istituto Italiano di Tecnologia, ADVR Advanced Robotics, 16163 Genoa, Italy
2   Dipartimento di Informatica, Bioingegneria, Robotica e Ingegneria dei Sistemi, Scuola Politecnica, Università Degli Studi Di Genova, 16145 Genoa, Italy
*   Correspondence: daniele.ludovico@iit.it
†   These authors contributed equally to this work.

**Abstract:** In recent years, the logistics sector expanded significantly, leading to the birth of smart warehouses. In this context, a key role is represented by autonomous mobile robots, whose main challenge is to find collision-free paths in their working environment in real-time. Model Predictive Control Algorithms combined with global path planners, such as the A* algorithm, show great potential in providing efficient navigation for collision avoidance problems. This paper proposes a Dual Forward–Backward Algorithm to find the solution to a Model Predictive Control problem in which the task of driving a mobile robotic platform into a bi-dimensional semi-structured environment is formulated in a convex optimisation framework.

**Keywords:** path planning for multiple mobile robots or agents; collision avoidance; optimization and optimal control

## 1. Introduction

In recent decades, the global economy has seen a massive development in the global value chain combined with the growth of "Industry 4.0" and "Internet plus" phenomena [1,2]. In spite of the impact of the COVID-19 pandemic, in an increasingly interconnected world, the flow of goods across different countries has become more and more frequent. This has led to a significant expansion of logistics markets [3]. With these developments, the need for greater efficiency and reliability in the transportation and storage process has become more and more intense. Against this background, the smart logistics industry, whose goal is the transition of the logistics industry to new technologies, has come into being. The smart warehouse is an automated warehouse based on several automated and interconnected technologies. It is characterised by high levels of automation, low labour costs, and high efficiency [4]. To develop a smart warehouse, a key role is represented by warehouse logistics robots. Indeed, in the process of rapid development of the logistics industry, smart sorting is one of the critical links to ensure that the logistics process can handle, in a timely manner, the massive material demand. Ultimately, the goal is to find a collision-free path between two points in the work environment. Implementing an optimal path planning algorithm would significantly reduce the operation time of logistics robots, while also reducing wear and energy consumption, and increasing the productivity and overall quality of the logistics industry. In addition, automation of such tasks would relieve human workers of repetitive and dull activities, which may cause harm to their health. Moreover, the COVID-19 pandemic makes the use of robots more attractive as they are not affected by health concerns or lock-down policies [5]. More generally, path planning of mobile robots in semi-structured environments, i.e., environments in which only the position of a subset

of obstacles is known a priori, is a crucial topic in robotics, not just for the logistics sector [6]. The pursuit of increasingly optimised models and algorithms in terms of path length and smoothness, running time, and reliability is an active research field.

Path planning for mobile robots is mainly classified into two categories: global planning and local planning [7]. From all the known techniques, Model Predictive Control Algorithms (MPCA), in combination with global path planning strategies such as A* algorithm [8], show some of the greatest potential to provide efficient navigation in collision avoidance problems [9]. In this context, a common strategy is to model obstacles as multiple convex polytopes leading to a convex optimisation problem whose solution is guaranteed [10].

Several algorithms have been proposed to solve convex optimisation problems. Several early approaches were based on the Active-Set (AS) method [11], which initially estimates the optimal active set. This algorithm then repeatedly uses gradient and Lagrange multiplier information to eliminate one index from the current estimate of the active constraints while adding a new index until optimality is reached [12]. Later, algorithms based on the Interior-Point (IP) method, which involves modelling the constraints as barrier functions [13], became popular in solving convex optimisation problems

The key novelty of this work is the development and use of a Dual Forward–Backward Algorithm (DFBA) [14] to solve a Model Predictive Control problem. This DFBA can compute the optimal trajectory to avoid obstacles in a convex optimisation framework, and this is vital for obstacle avoidance in mobile robotic platforms operating in two-dimensional, semi-structured environments such as warehouses. Although the DFBA requires a longer computational time than the IP and AS algorithms, formulation of the optimisation problem associated with MPCA is relatively straightforward, and this is suited to real-time applications.

Section 2 presents the physical model of the system and the geometrical representation of the environment. This is then used to define the mathematical formulation in terms of an optimisation problem. Finally, there is a description of the algorithms needed to solve the proposed optimisation problem. A case study to test the proposed algorithms and the corresponding results are described in Section 3. A discussion of the results is presented in Section 4. Finally, the conclusion and future works are presented in Section 5.

## 2. Materials and Methods

This section presents a mathematical description of the problem introduced in Section 1, defines its equivalent optimisation problem, and describes the proposed numerical algorithms to solve it.

### 2.1. Problem Description

This subsection describes the system and environment models employed to solve the problem of autonomously driving a mobile robotic platform, avoiding obstacles in a two-dimensional, semi-structured environment.

#### 2.1.1. Dynamic Model

As a first approximation, the mobile robot can be modelled as a point mass moving on a plane without considering gravity effects. The corresponding equation of motion is

$$\ddot{p} = u,$$

where $\ddot{p}$ is the acceleration of the robot and $u$ is the actuator control signal. The improved Euler method [15] can be used to obtain the discrete time model of the system:

$$\begin{cases} v_{k+1} = v_k + u_k T_s \\ p_{k+1} = p_k + v_k T_s + \frac{u_k}{2} T_s^2 \end{cases}, \tag{1}$$

where $T_s$ is the sample time, $\boldsymbol{u}_k$ is the actuator control signal at a given time instant $k$, while $\boldsymbol{p}_{k+1}$, $\boldsymbol{v}_{k+1}$ and $\boldsymbol{p}_k$, $\boldsymbol{v}_k$ are the position and velocity at a given time instant $k+1$ and $k$, respectively. Equation (1) can be written in state-space form as follows:

$$z_{k+1} = A z_k + B u_k, \tag{2}$$

where $\boldsymbol{z}_k = \begin{bmatrix} \boldsymbol{p}_k & \boldsymbol{v}_k \end{bmatrix}^T$ and $\boldsymbol{z}_{k+1} = \begin{bmatrix} \boldsymbol{p}_{k+1} & \boldsymbol{v}_{k+1} \end{bmatrix}^T$ are the states of the system at a given time instant $k$ and $k+1$, respectively. Details about matrices $A$ and $B$ are presented in Appendix A. In the problem presented here, dynamic laws are considered as equality constraints. Indeed, (2) can be rearranged as follows:

$$- A z_k - B u_k + \mathbf{I}_4 z_{k+1} = \mathbf{0}_{4,1}, \tag{3}$$

where $\mathbf{I}_n$ is the $n \times n$ identity matrix and $\mathbf{0}_{m,n}$ is the $m \times n$ zero matrix.

### 2.1.2. Actuation Model

A real actuator is not able to produce arbitrarily large accelerations and velocities. This saturation phenomenon can be modelled as the set, $\Delta$, defined by the intersection of linear inequalities:

$$\Delta = \left\{ \begin{array}{l} \forall \boldsymbol{u}_k \in \mathbb{R}^2, \forall \boldsymbol{v}_k \in \mathbb{R}^2 : \\ -u_{min}\, \mathbb{1}_{2,1} \leq \boldsymbol{u}_k \leq u_{max}\, \mathbb{1}_{2,1} \\ -v_{min}\, \mathbb{1}_{2,1} \leq \boldsymbol{v}_k \leq v_{max}\, \mathbb{1}_{2,1} \end{array} \right\}, \tag{4}$$

where $\mathbb{1}_{m,n}$ is a $m \times n$ matrix all of whose entries are one, while $u_{max}$, $u_{min}$, $v_{max}$, and $v_{min}$ are real positive numbers representing the limits of the actuator.

### 2.1.3. Obstacle Model

All the obstacles are modelled as circular or rectangular subsets of $\mathbb{R}^2$, as reported in Figure 1. A generic circular obstacle $O_{c_i}$ is defined as follows:

$$O_{c_i} = \left\{ \begin{array}{l} \forall \boldsymbol{P} \in \mathbb{R}^2, \boldsymbol{P}_{c_i} \in \mathbb{R}^2, \rho_i \in \mathbb{R} : \\ \|\boldsymbol{P} - \boldsymbol{P}_{c_i}\|^2 < \rho_i^2 \end{array} \right\},$$

where $\boldsymbol{P}_{c_i}$ and $\rho_i$ are the centre and the radius of $O_{c_i}$, respectively. Similarly, a generic rectangular obstacle $O_{r_i}$ is defined as follows:

$$O_{r_i} = \left\{ \begin{array}{l} \forall \boldsymbol{P} \in \mathbb{R}^2, \boldsymbol{r}_{i_j} \in \mathbb{R}^2, \boldsymbol{V}_{i_j} \in \mathbb{R}^2 : \\ \left\langle \boldsymbol{r}_{i_j}, \boldsymbol{P} - \boldsymbol{V}_{i_j} \right\rangle < 0 \text{ for } j = 1, \ldots, 4 \end{array} \right\},$$

where $\boldsymbol{V}_{i_j}$ is the $j^{th}$ vertex of $O_{r_i}$, $\boldsymbol{r}_{i_j}$ is the outward-pointing normal vector to the $j^{th}$ side of $O_{r_i}$, and $\langle \cdot, \cdot \rangle$ indicates the inner product. Then, the robot is constrained to move into a non-convex set:

$$\Pi = \mathbb{R}^2 \setminus \{O_c \cup O_r\},$$

where

$$O_c = \left\{ \bigcup_{i=1}^{n_c} O_{c_i} \right\}, \quad O_r = \left\{ \bigcup_{i=1}^{n_r} O_{r_i} \right\},$$

and $n_c$ and $n_r$ are the number of circular and rectangular obstacles, respectively.

The maximum convex polytope $\Omega_t \subset \Pi$ containing the centre of the robot $\boldsymbol{p}_k$, as shown in Figure 2, is defined as

$$\Omega_t = \left\{ \begin{array}{l} \forall \boldsymbol{P} \in \mathbb{R}^2, \boldsymbol{n}_i \in \mathbb{R}^2, \boldsymbol{P}_i \in \mathbb{R}^2 : \\ -\langle \boldsymbol{n}_i, \boldsymbol{P} - \boldsymbol{P}_i \rangle \leq 0 \text{ for } i = 1, \ldots, n_a \end{array} \right\}, \tag{5}$$

where $n_a$ is the minimum number of lines necessary to define $\Omega_t$, while $\boldsymbol{n}_i$ and $\boldsymbol{P}_i$ have different meaning depending on the obstacle type. For circular obstacles, $\boldsymbol{n}_i = \boldsymbol{p}_k - \boldsymbol{P}_{c_i}$ and $\boldsymbol{P}_i$ is the point on the boundary of the $i^{th}$ circular obstacle at minimum distance from $\boldsymbol{p}_k$. For each rectangular obstacle, the eight sets $\Theta_{i_j}$ and $\Xi_{i_j}$ with $j = 1, \ldots, 4$ are defined, as shown in Figure 1b. If $\boldsymbol{p}_k \in \Theta_{i_j}$, $\boldsymbol{n}_i = \boldsymbol{r}_{i_j}$ and $\boldsymbol{P}_i = \boldsymbol{V}_{i_j}$, otherwise, if $\boldsymbol{p}_k \in \Xi_{i_j}$, $\boldsymbol{n}_i = \boldsymbol{p}_k - \boldsymbol{V}_{i_j}$ and $\boldsymbol{P}_i = \boldsymbol{V}_{i_j}$.

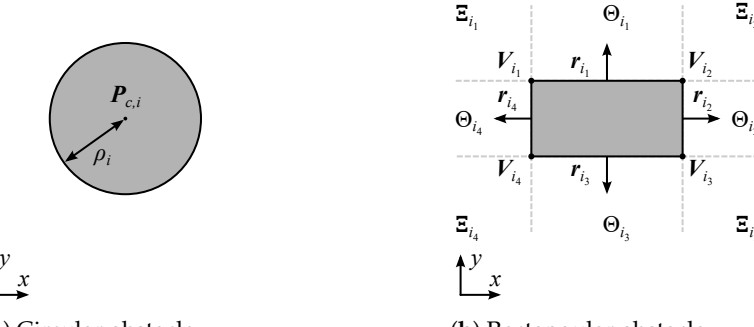

(**a**) Circular obstacle

(**b**) Rectangular obstacle.

**Figure 1.** Obstacles model.

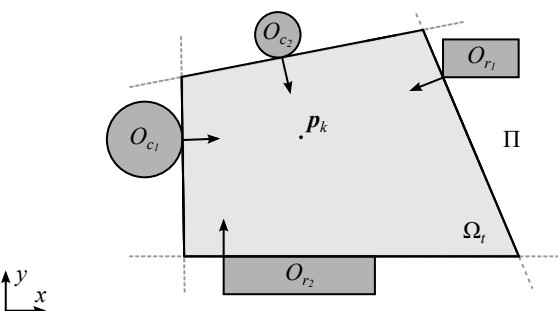

**Figure 2.** Example of computing the maximum convex polytope $\Omega_t$.

### 2.2. Optimisation Problem

The problem described in Section 2.1 can be formulated in a convex optimisation framework. The objective is to find the optimal control inputs to autonomously drive a mobile robot from an initial position to a target position within a semi-structured environment avoiding obstacles.

#### 2.2.1. Objective Function

Considering $n_p$ predictions of the system states and control variables, it is possible to define the optimisation variables $\boldsymbol{\xi} \in \mathbb{R}^{n_v}$, where $n_v = 6n_p + 4$, as

$$\boldsymbol{\xi} = \begin{bmatrix} \boldsymbol{z}_0^T & \boldsymbol{z}_1^T & \cdots & \boldsymbol{z}_{n_p}^T & \boldsymbol{u}_1^T & \cdots & \boldsymbol{u}_{n_p}^T \end{bmatrix}^T.$$

Then, introducing the vector containing the initial state, the $n_p$ desired states, and the $n_p$ desired control inputs (set to zero to reduce the control effort)

$$\boldsymbol{\xi}_d = \begin{bmatrix} \boldsymbol{z}_{d_0}^T & \boldsymbol{z}_{d_1}^T & \cdots & \boldsymbol{z}_{d_{n_p}}^T & \boldsymbol{0}_{1,2} & \cdots & \boldsymbol{0}_{1,2} \end{bmatrix}^T,$$

the objective can be modelled as a quadratic function

$$f(\boldsymbol{\xi}) = \frac{1}{2} \|Q(\boldsymbol{\xi} - \boldsymbol{\xi}_d)\|_2^2, \tag{6}$$

where the matrix $Q \in \mathbb{R}^{n_v \times n_v}$ is diagonal and allows weighting the cost function to define the priority between increasing the trajectory tracking accuracy in terms of position and velocity and reducing the control effort. The matrix $Q$ has the following structure:

$$Q = \mathrm{diag}\left(\begin{bmatrix} w_z & \dots & w_z & w_u & \dots & w_u \end{bmatrix}\right),$$

where the weights of the position error and the velocity error

$$w_z = \begin{bmatrix} w_p & w_p & w_v & w_v \end{bmatrix}$$

are repeated $n_p + 1$ times, while the weight of the control effort

$$w_u = \begin{bmatrix} w_u & w_u \end{bmatrix}$$

is repeated $n_p$ times. The parameters $w_p$, $w_v$, and $w_u$ are real positive numbers.

### 2.2.2. Equality Constraints

Equation (3) can be seen as an equality constraint which forces the states of the system at instant $k + 1$ to satisfy the dynamic equations. Considering a prediction horizon of $n_p$ samples, it is possible to write (3) at each $T_s$ in terms of the optimisation variables $\xi$ as linear equality constraints

$$\Phi\xi - b = 0_{4(n_p+1),1}. \tag{7}$$

Details about matrix $\Phi$ and vector $b$ are presented in Appendix A.

### 2.2.3. Inequality Constraints

As described in Section 2.1, both the limits of the actuators and the region in which the robotic platform can move are modelled as linear inequalities. Considering a prediction horizon of $n_p$ samples, it is possible to write (4) and (5) in terms of the optimisation variables and in compact form as

$$C\xi - d \le 0_{n_d,1}, \tag{8}$$

where $n_d = n_p(n_a + 8) + n_a + 4$. Details about matrix $C$ and vector $d$ are presented in Appendix B.

### 2.2.4. Dual Problem Formulation

Considering the objective function (6) and the constraints (7) and (8), it is possible to define the following optimisation problem:

$$\begin{aligned}
\min_{\xi \in \mathbb{R}^{n_v}} \quad & \frac{1}{2}\|Q(\xi - \xi_d)\|_2^2 \\
\text{subject to} \quad & \Phi\xi - b = 0_{4(n_p+1),1}. \\
& C\xi - d \le 0_{n_d,1}
\end{aligned} \tag{9}$$

Since the objective function is quadratic and the constraints are linear, this optimisation problem is convex. Assuming that a solution to (9) exists, since the objective function is quadratic, it is also continuous at some $\xi$ such that constraints (7) and (8) are satisfied. Assuming $x \in \mathbb{R}^n$ and the subset $\Gamma \in \mathbb{R}^n$, where $n \in \mathbb{N}$, the indicator function of $\Gamma$ can be defined as

$$\iota_\Gamma(x) = \begin{cases} 0 & x \in \Gamma \\ +\infty & x \notin \Gamma \end{cases}.$$

Then, introducing the linear operator

$$L = \begin{bmatrix} \Phi^T & C^T \end{bmatrix}^T$$

and the vector

$$\boldsymbol{\beta} = \begin{bmatrix} \boldsymbol{b}^T & \boldsymbol{d}^T \end{bmatrix}^T,$$

the optimisation problem described in (9) can be equivalently rewritten as

$$\min_{\boldsymbol{\xi} \in \mathbb{R}^{n_v}} \quad \frac{1}{2} \|Q(\boldsymbol{\xi} - \boldsymbol{\xi}_d)\|_2^2 + \iota_F(L\boldsymbol{\xi} - \boldsymbol{\beta}), \tag{10}$$

where $F$ is a set defined as follows:

$$F = \left\{ \begin{array}{l} \boldsymbol{\lambda} \in \mathbb{R}^{4(n_p+1)+n_d} : \\ \lambda_i = 0 \text{ for } i = 1, \dots, 4(n_p + 1), \\ \lambda_i \leq 0 \text{ for } i = 4(n_p + 1) + 1, \dots, 4(n_p + 1) + n_d \end{array} \right\}.$$

Defining

$$g(\boldsymbol{\xi}) = \iota_F(\boldsymbol{\xi} - \boldsymbol{\beta}),$$

it is possible to write (10) as

$$\min_{\boldsymbol{\xi} \in \mathbb{R}^{n_v}} \quad f(\boldsymbol{\xi}) + g(L\boldsymbol{\xi}). \tag{11}$$

The convex conjugate [16] of (11) has the following form:

$$\min_{\boldsymbol{\lambda} \in \mathbb{R}^{4(n_p+1)+n_d}} \quad f^*\left(-L^T\boldsymbol{\lambda}\right) + g^*(\boldsymbol{\lambda}) + \langle \boldsymbol{\lambda}, \boldsymbol{\beta} \rangle,$$

where $f^*(\boldsymbol{\zeta})$, $\boldsymbol{\zeta} \in \mathbb{R}^{n_v}$, is the Legendre-Fenchel conjugate of $f(\boldsymbol{\xi})$ and is equal to

$$f^*(\boldsymbol{\zeta}) = \frac{1}{2}\left\|Q^{-1}\boldsymbol{\zeta}\right\|_2^2 + \langle \boldsymbol{\xi}_d, \boldsymbol{\zeta} \rangle,$$

while $g^*(\boldsymbol{\lambda})$ is the Legendre-Fenchel conjugate of $g(\boldsymbol{\xi})$ and is equal to

$$g^*(\boldsymbol{\lambda}) = \iota_{F^\circ}(\boldsymbol{\lambda}),$$

where

$$F^\circ = \left\{ \boldsymbol{\lambda} \in \mathbb{R}^{4(n_p+1)+n_d} : \lambda_i \geq 0 \text{ for } i = 4(n_p + 1) + 1, \dots, 4(n_p + 1) + n_d \right\}.$$

### 2.3. Algorithms

This section presents the algorithms needed to solve the optimisation problem described in Section 2.2. First, the MPCA is explained in detail. Then, the Convex Subset Search Algorithm (CSSA), which is employed to compute the maximum convex subset in which the mobile robot can move avoiding the obstacles, is presented. Finally, the DFBA used to solve (9) is described.

#### 2.3.1. Model Predictive Control Algorithm

Exploiting the current and desired states of the system and the obstacles map, the MPCA computes the optimal sequence of inputs to obtain the $n_p$ predicted states that minimise the objective function. Then, only the first input is applied and all the variables of the optimisation problem (the obstacles map, the constraints matrices $C$ and vectors $\boldsymbol{d}$, and $\boldsymbol{b}$) are updated according to the newly measured position. Moreover, to reduce the number of iterations required by the DFBA, the initial point of the optimisation solver, $\boldsymbol{\xi}_i$, is updated using the predictions of the states of the system obtained as the results of the optimisation problem at the previous time instant. The algorithm is repeated every sample time $T_s$. For this application, to define the desired states, the trajectory that connects the initial position to the target one is computed using the A* algorithm. The pseudocode of the MPCA is described in Algorithm 1.

---

**Algorithm 1** MPCA

---

1: $k = 0$
2: $\boldsymbol{p}_k \leftarrow$ *initial position*
3: $\boldsymbol{p}_f \leftarrow$ *target position*
4: $\boldsymbol{p}_t \leftarrow$ *compute trajectory via $A^*$ algorithm*
5: $map \leftarrow$ *obstacles map*
6: $\Omega_t \leftarrow CSSA(\boldsymbol{p}_k, map)$
7: $u_{max}$ , $u_{min}$ $\leftarrow$ *acceleration actuators limits*
8: $v_{max}$ , $v_{min}$ $\leftarrow$ *velocity actuators limits*
9: $C$ , $\boldsymbol{d}$ $\leftarrow$ *compute inequality constraint parameters*
10: $n_p \leftarrow$ *prediction horizon*
11: $T_s \leftarrow$ *sample time*
12: $\Phi$ , $\boldsymbol{b}$ $\leftarrow$ *compute equality constraint parameters*
13: $Q \leftarrow$ *define weight matrix*
14: $\mathfrak{Q} = \left(Q^T Q\right)^{-1}$
15: $\boldsymbol{\xi}_i = \boldsymbol{0}_{6n_p+4,1} \leftarrow$ *initial point of DFBA*
16: $\delta \leftarrow$ *position tolerance*
17: **while** $\left\| \boldsymbol{p}_f - \boldsymbol{p}_k \right\|_2 < \delta$ **do**
18:     $\boldsymbol{\xi}_d \leftarrow$ *update with $\boldsymbol{p}_t$*
19:     $\boldsymbol{\xi}_k \leftarrow DFBA(\boldsymbol{\xi}_i, \boldsymbol{\xi}_d, \mathfrak{Q}, \Phi, \boldsymbol{b}, C, \boldsymbol{d})$
20:     $\boldsymbol{u}_k \leftarrow$ *apply $\boldsymbol{u}_0$ in vector $\boldsymbol{\xi}_k$*
21:     $\boldsymbol{p}_k \leftarrow$ *update current position with measurement*
22:     $map \leftarrow$ *update obstacles map*
23:     $\Omega_t \leftarrow CSSA(\boldsymbol{p}_k, map)$
24:     $C$ , $\boldsymbol{d}$ $\leftarrow$ *update inequality constraints*
25:     $\boldsymbol{b} \leftarrow$ *update dynamic constraints with $\boldsymbol{p}_k$*
26:     $\boldsymbol{\xi}_i \leftarrow$ *update initial point of DFBA*
27:     $k = k + 1$
28: **end while**

---

### 2.3.2. Convex Subset Search Algorithm

The CSSA takes as its input the current robot position and the updated map describing the obstacles and returns the minimum number of vectors, $\{\boldsymbol{n}_1, \ldots, \boldsymbol{n}_{n_a}\}$ and $\{\boldsymbol{P}_1, \ldots, \boldsymbol{P}_{n_a}\}$, to compute the inequality constraints required to define $\Omega_t$. The pseudocode of the CSSA is described in Algorithm 2.

### 2.3.3. Dual Forward–Backward Algorithm

Since $f$ is strongly convex with a modulus of convexity $\mu > 0$, $f^*$ is differentiable on $F^o$, and $\nabla f^*$ is $1/\mu$-Lipschitz continuous, the primal problem has a unique solution $\hat{\boldsymbol{\xi}}$. Furthermore, assuming that the calculus rule for subdifferentials holds, then the dual solution $\hat{\boldsymbol{\lambda}}$ also exists, the duality gap is zero, and the following Karush–Kuhn–Tucker conditions [13] hold:

$$\hat{\boldsymbol{\xi}} = \nabla f^*\left(-L^* \hat{\boldsymbol{\lambda}}\right) \quad \text{and} \quad L\hat{\boldsymbol{\xi}} \in \partial g^*\left(\hat{\boldsymbol{\lambda}}\right).$$

---

**Algorithm 2** CSSA$(p_k, map)$

---

1: $h = 0$
2: **while** isEmpty$(map)$ **do**
3:     $O_i \leftarrow$ *obstacle at minimum distance from* $p_k$
4:     $i \leftarrow$ *index of* $O_i$
5:     **if** isRectangle$(O_i)$ **then**
6:        **if** $p_k \in \Theta_{i_j}$ **then**
7:           $n_h = r_{i_j}$
8:        **else**
9:           $n_h = p_k - V_{i_j}$
10:        **end if**
11:        $P_h = V_{i_j}$
12:     **else**
13:        $n_h = p_k - P_{c_i}$
14:        $P_h = P_{c_i} + \rho_i n_h / \|n_h\|_2$
15:     **end if**
16:     $t = 0$
17:     $map \leftarrow$ *update by removing* $O_i$
18:     **for** $q = 0$ to length$(map) - 1$ **do**
19:        **if** isDisk$(O_q)$ **then**
20:           $P_{md} = P_{c_q} + \rho_q n_h / \|n_h\|_2$
21:           **if** $\langle n_h, P_{md} - n_h \rangle < 0$ **then**
22:              $R_t = O_q$
23:              $t = t + 1$
24:           **end if**
25:        **else**
26:           $bool = true$
27:           **for** $m = 1$ to 4 **do**
28:              $bool = bool \wedge \left( \langle n_h, V_{q_m} - P_h \rangle < 0 \right)$
29:           **end for**
30:           **if** $bool$ **then**
31:              $R_t = O_q$
32:              $t = t + 1$
33:           **end if**
34:        **end if**
35:     **end for**
36:     $map \leftarrow$ *update by removing* $R$
37:     $h = h + 1$
38: **end while**

---

Thus, the dual solution uniquely determines the primal solution. Under these conditions, it is possible to solve (9) using the following DFBA:

$$\xi_h = \nabla f^*(-L^* \lambda_h),$$
$$\lambda_{h+1} = \text{prox}_{\gamma g^*}(\lambda_h + \gamma L \xi_h),$$
$$\lambda_{h+1} = \text{prox}_{F^o}(\lambda_{h+1}),$$

where

$$\nabla f^*(-L^* \lambda_h) = -\left( Q^T Q \right)^{-1} \left( L^T \lambda_h \right) + \xi_d,$$
$$\text{prox}_{\gamma g^*}(\lambda_h + \gamma L \xi_h) = \lambda_h + \gamma(L\xi - \beta),$$
$$\text{prox}_{F^o}(\lambda_h) = \begin{cases} \lambda_{h,i} & \lambda_{h,i} \geq 0 \vee i \leq 4(n_p + 1) \\ 0 & \lambda_{h,i} < 0 \wedge i > 4(n_p + 1) \end{cases},$$

$\lambda_{h,i}$ is the $i$th component of the optimisation variable of the dual problem at the $h$th iteration, and $0 < \gamma < \gamma_m$ with $\gamma_m = 2\mu / \|L\|^2$, where the modulus of convexity $\mu$ is computed as the minimum eigenvalue of $Q^T Q$. The pseudocode of the DFBA is described in Algorithm 3.

---

**Algorithm 3** DFBA$(\boldsymbol{\xi}_i, \boldsymbol{\xi}_d, \mathfrak{Q}, \Phi, \boldsymbol{b}, C, \boldsymbol{d})$

---

1:  $\boldsymbol{L} \leftarrow$ *generated by* $\Phi$ *and* $C$
2:  $\beta \leftarrow$ *generated by* $\boldsymbol{b}$ *and* $\boldsymbol{d}$
3:  $\boldsymbol{\lambda}_0 = \boldsymbol{0}_{4(n_p+1)+n_d,1}$
4:  $\boldsymbol{\xi}_0 = \boldsymbol{\xi}_i$
5:  $\gamma \leftarrow$ *initialise considering* $L$
6:  $tol \leftarrow$ *DFBA termination tolerance*
7:  $maxIter \leftarrow$ *DFBA maximum number of iterations*
8:  **for** $h = 0$ to $maxIter - 1$ **do**
9:    $\boldsymbol{\lambda}_{h+1} = \boldsymbol{\lambda}_h + \gamma(L\boldsymbol{\xi}_h - \beta)$
10:   **for** $i = 4(n_p + 1) + 1$ to $\text{length}(\boldsymbol{\lambda}_{h+1})$ **do**
11:     **if** $\lambda_{h+1,i} \le 0$ **then**
12:      $\lambda_{h+1,i} = 0$
13:     **end if**
14:   **end for**
15:   $\boldsymbol{\xi}_{h+1} = \mathfrak{Q}\left(-\boldsymbol{L}^T\boldsymbol{\lambda}_{h+1}\right) + \boldsymbol{\xi}_d$
16:   **if** $\left\| \boldsymbol{\xi}_{h+1} - \boldsymbol{\xi}_h \right\|_2 / \text{length}(\boldsymbol{\xi}_h) < tol$ **then**
17:    *stop iteration*
18:   **end if**
19: **end for**

---

## 3. Results

A case study was developed to test the algorithms described in Section 2.3. Consider a $48\,\text{m} \times 36\,\text{m}$ warehouse that contains eight shelves and three mobile robots ($R_{m_1}$, $R_{m_2}$, and $R_{m_3}$). The task for each robot is to reach different target positions while avoiding the walls, the shelves, and the other robots. As described in Section 2.1, the walls and the shelves are modelled as rectangular obstacles, whose parameters are reported in Table 1, while the mobile robots are modelled as circular obstacles with a diameter equal to 1 m.

**Table 1.** Rectangular Obstacles Parameters.

| | $V_{i,1}$ | | $V_{i,2}$ | | $V_{i,3}$ | | $V_{i,4}$ | | Unit |
|---|---|---|---|---|---|---|---|---|---|
| | $x$ | $y$ | $x$ | $y$ | $x$ | $y$ | $x$ | $y$ | |
| $O_1$ | 6 | 32 | 22 | 32 | 22 | 30 | 6 | 30 | m |
| $O_2$ | 28 | 32 | 44 | 32 | 44 | 30 | 28 | 30 | m |
| $O_3$ | 6 | 24 | 22 | 24 | 22 | 22 | 6 | 22 | m |
| $O_4$ | 28 | 24 | 44 | 24 | 44 | 22 | 28 | 22 | m |
| $O_5$ | 6 | 16 | 22 | 16 | 22 | 14 | 6 | 14 | m |
| $O_6$ | 28 | 16 | 44 | 16 | 44 | 14 | 28 | 14 | m |
| $O_7$ | 6 | 8 | 22 | 8 | 22 | 6 | 6 | 6 | m |
| $O_8$ | 28 | 8 | 44 | 8 | 44 | 6 | 28 | 6 | m |
| $O_9$ | 0 | 38 | 1 | 38 | 1 | 0 | 0 | 0 | m |
| $O_{10}$ | 49 | 38 | 50 | 38 | 50 | 0 | 49 | 0 | m |
| $O_{11}$ | 1 | 38 | 49 | 38 | 49 | 37 | 1 | 37 | m |
| $O_{12}$ | 1 | 1 | 49 | 1 | 49 | 0 | 1 | 0 | m |

Since all the constraints of the optimisation problem are computed considering each mobile robot to be a point mass, all the obstacles are enlarged by the radius of the mobile robot. Given this environment, the following logistic task is simulated: (i) $R_{m_1}$ has to move an object from point $X$ to point $Y$, then returns to its home position $H_1$; (ii) $R_{m_2}$ has to move an object from point $Y$ to point $X$, then returns to its home position $H_2$; (iii) $R_{m_3}$ has to store

an object at point **Z**, then returns to the home position **H**$_3$. The $x$ and $y$ coordinates of the above-mentioned target positions are listed in Table 2.

**Table 2.** Target Positions.

|   | $H_1$ | $H_2$ | $H_3$ | $X$ | $Y$ | $Z$ | Unit |
|---|-------|-------|-------|-----|-----|-----|------|
| $x$ | 3 | 5 | 7 | 14 | 32 | 40 | m |
| $y$ | 36 | 36 | 36 | 10 | 20 | 10 | m |

Figure 3 shows the simulation environment where the following elements are associated with each robot: (i) a disk representing the robot itself; (ii) a polytope representing the allowed area in which the robot can move; (iii) the path travelled by the robot; (iv) the current target position labelled with the × symbol; (v) the home position represented as a dot. The elements associated with $R_{m_1}$, $R_{m_2}$, and $R_{m_3}$ are depicted in blue, red, and yellow, respectively.

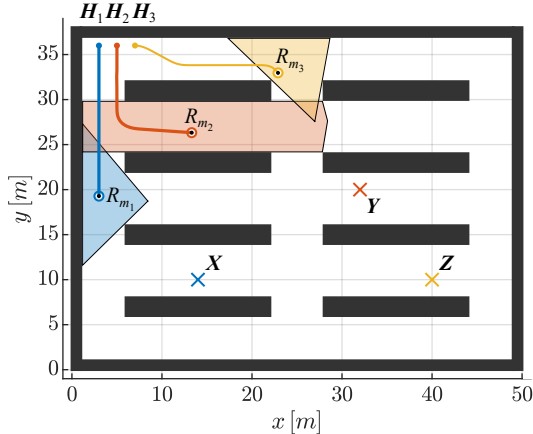

**Figure 3.** A frame of the simulation of the logistic task showing the mobile robots with their allowed area, their travelled path, and their current target.

A simulation, whose parameters are reported in Table 3, was implemented in Matlab® 2019b using the Parallel Computing Toolbox™ to measure the performances of the proposed algorithm.

**Table 3.** Simulation Parameters.

| Algorithm | Parameter | Value | Unit |
|-----------|-----------|-------|------|
|           | $\delta$ | 0.1 | m |
|           | $w_p$ | 5 | 1/m |
|           | $w_v$ | 3 | s/m |
|           | $w_u$ | 1 | s$^2$/m |
| MPCA      | $u_{max}$ | 5 | m/s$^2$ |
|           | $u_{min}$ | 5 | m/s$^2$ |
|           | $v_{max}$ | 1.5 | m/s |
|           | $v_{min}$ | 1.5 | m/s |
|           | $T_s$ | 0.1 | s |
|           | $n_p$ | 10 | |
|           | $maxIter$ | $5 \times 10^4$ | |
| DFBA      | $tol$ | $1 \times 10^{-6}$ | |
|           | $\mu$ | 1 | |
|           | $\gamma$ | $0.99\,\gamma_m$ | |

The weights were chosen to promote the trajectory tracking performances rather than reducing the control effort. Figure 4 shows how the proposed algorithm avoids a crash between $R_{m_1}$ and $R_{m_2}$. In Figure 4a, since the desired trajectory of each robot lies within the allowed area, the mutual obstacle constraints are inactive. In Figure 4b, the predicted positions deviate from the desired paths to satisfy the obstacle constraints. In Figure 4c, the two robots react to avoid a crash, leading to an increase in the trajectory tracking error. In Figure 4d, the two robots start to recover their desired paths.

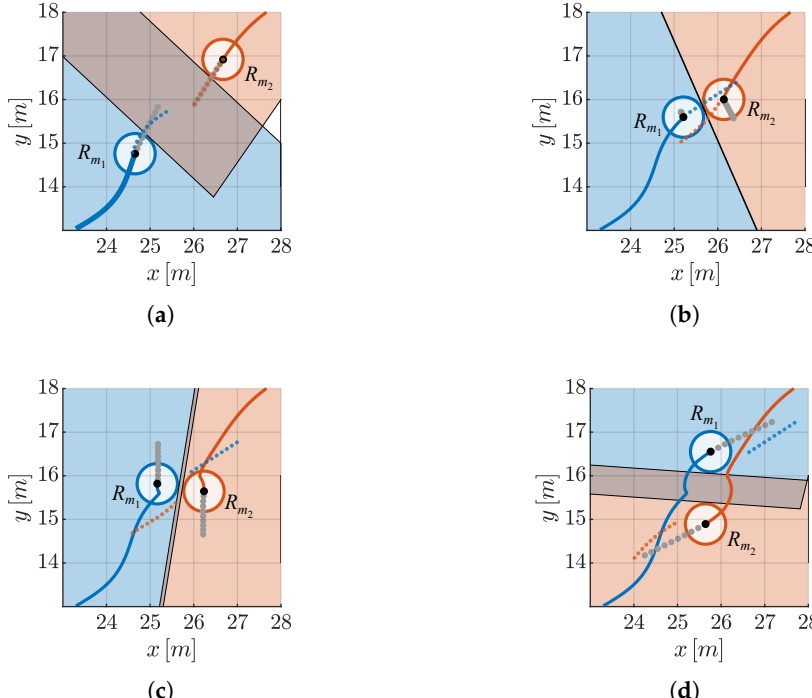

**Figure 4.** Detail of how the proposed algorithm avoids a crash between $R_{m_1}$ and $R_{m_2}$. For each robot, the predicted positions (grey dots), the desired trajectory (coloured dots), the travelled path (coloured line), and the allowed area (coloured polytope) are reported. $R_{m_1}$ and $R_{m_2}$ are shown in blue and red, respectively. (**a**) The desired trajectory of each robot lies within the allowed area, hence the obstacles constraints are inactive; (**b**) The predicted positions deviate from the desired paths to satisfy the obstacle constraints; (**c**) The two robots react to avoid a crash; (**d**) The two robots start to recover their desired paths.

The results of the simulation show that all the constraints of the optimisation problem are satisfied and the trajectory tracking error is bounded, except when the constraints are active. Figure 5 summarises the simulation results. In particular, Figure 5a shows the trend over time of the norm of the trajectory tracking error, Figure 5b represents the maximum value of the constraint of the nearest obstacle, while the velocity and control effort of each mobile robot along the $x$-axis and $y$-axis are depicted in Figure 5c–e, respectively.

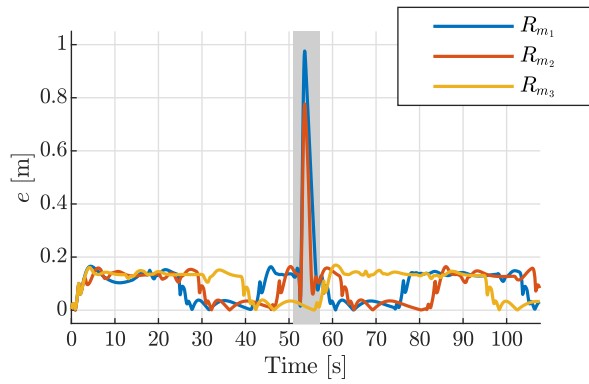

(**a**) Norm of the trajectory tracking error.

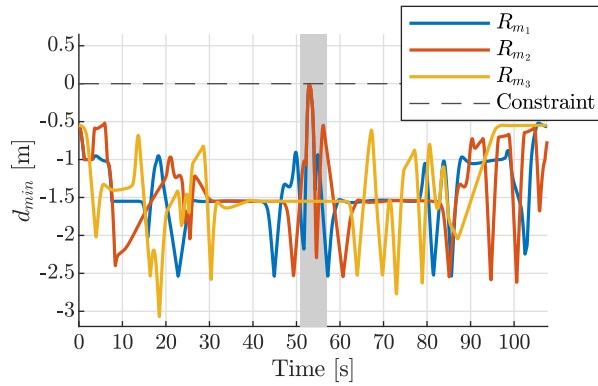

(**b**) Value of the constraint of the nearest obstacle.

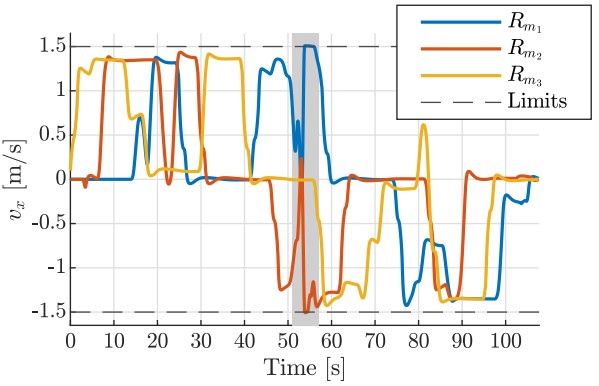

(**c**) Velocity along the $x-$axis.

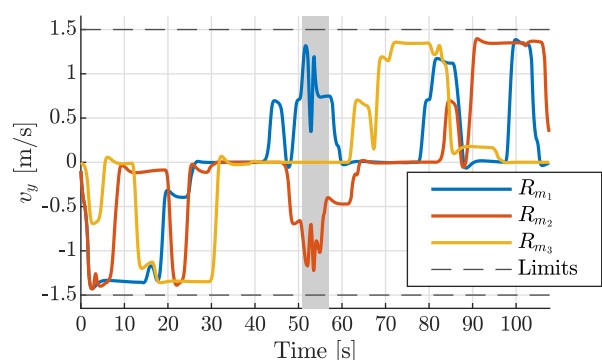

(**d**) Velocity along the $y-$axis.

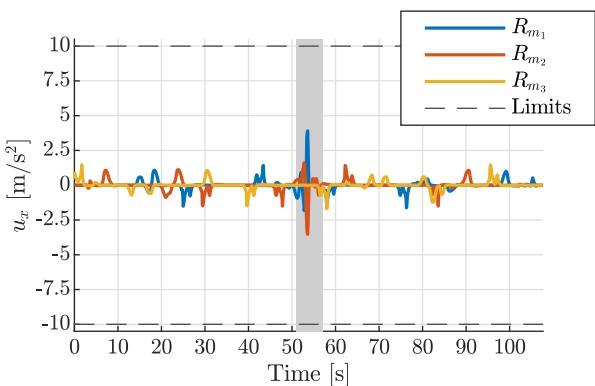

(**e**) Control effort along the $x-$axis.

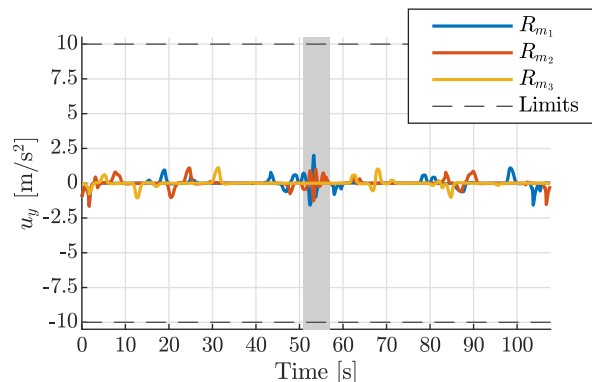

(**f**) Control effort along the $y-$axis.

**Figure 5.** Simulation results. The shaded areas correspond to the avoided collision event described in Figure 4.

## 4. Discussion

The performance of the proposed DFBA is evaluated by comparing its results with those obtained using IP and AS algorithms to solving the optimisation problem presented in the case study in Section 3. Both these algorithms are implemented in Matlab® 2019b using the standard functions of the Optimization Toolbox™, setting the maximum number of iterations and the termination tolerance equal to that of the DFBA, as reported in Table 3. The results of the simulations are presented in Table 4, where:

- $\hat{e}$ is the mean of the norm of the trajectory tracking error;
- $\sigma(e)$ is the sample standard deviation of the norm of the trajectory tracking error;
- $\max(d_{min})$ is the maximum value of the constraint of the nearest obstacle;

- $\max|v_x|$ is the maximum of the velocity of each mobile robot along the *x*-axis;
- $\max|v_y|$ is the maximum of the velocity of each mobile robot along the *y*-axis;
- $\max|u_x|$ is the maximum of control effort of each mobile robot along the *x*-axis.
- $\max|u_y|$ is the maximum of control effort of each mobile robot along the *y*-axis;
- $\hat{t}_c$ is the mean of the computation time required by the algorithm.

**Table 4.** Comparison of the results obtained using different optimisation algorithms.

| Parameter | $R_{m_1}$ | | | $R_{m_2}$ | | | $R_{m_3}$ | | | Unit |
|---|---|---|---|---|---|---|---|---|---|---|
| | DFBA | IP | AS | DFBA | IP | AS | DFBA | IP | AS | |
| $\hat{e}$ | 0.12 | 0.12 | 0.12 | 0.12 | 0.11 | 0.11 | 0.12 | 0.11 | 0.11 | m |
| $\sigma(e)$ | 0.04 | 0.03 | 0.03 | 0.03 | 0.03 | 0.03 | 0.04 | 0.04 | 0.04 | m |
| $\max(d_{min})$ | −0.023 | −0.052 | −0.035 | −0.011 | −0.048 | −0.031 | −0.547 | −0.545 | −0.545 | m |
| $\max|v_x|$ | 1.50 | 1.50 | 1.50 | 1.50 | 1.50 | 1.50 | 1.43 | 1.41 | 1.41 | m/s |
| $\max|v_y|$ | 1.43 | 1.40 | 1.40 | 1.43 | 1.40 | 1.40 | 1.36 | 1.36 | 1.36 | m/s$^2$ |
| $\max|u_x|$ | 3.91 | 5.00 | 3.81 | 3.53 | 4.77 | 3.53 | 1.69 | 1.46 | 1.46 | m/s |
| $\max|u_y|$ | 2.00 | 1.86 | 2.04 | 1.65 | 1.45 | 3.45 | 1.10 | 1.00 | 1.00 | m/s$^2$ |
| $\hat{t}_c$ | 0.030 | 0.022 | 0.025 | 0.029 | 0.021 | 0.024 | 0.027 | 0.022 | 0.022 | s |

All the tested algorithms are able to solve the optimisation problem resulting from driving a mobile robotic platform into a two-dimensional semi-structured environment avoiding obstacles autonomously. They have comparable accuracy regarding the trajectory tracking error and the capability of satisfying both the obstacles' constraints and the limits of the actuators. Concerning the computation time required to solve the optimisation problem, the IP and AS algorithms are, on average, 24.4% and 17.4% faster than the DFBA. However, unlike IP and AS algorithms, the DFBA can solve the Model Predictive Control problem using only matrix multiplications and operations, such as conditional statements and *for* loops. This feature is advantageous because it allows an efficient implementation of the MPCA in common micro-controllers.

## 5. Conclusions

In this paper, to address the problem of driving a mobile robotic platform in a two-dimensional semi-structured environment while autonomously avoiding obstacles, a Dual Forward–Backward Algorithm is proposed that can solve a Model Predictive Control Algorithm within a convex optimisation framework. First, the mathematical details about the problem formulation are described in terms of dynamic equations, system modelling, and algorithms. Then, a case study from the logistics sector, in which three mobile robots have to reach different target positions avoiding walls, shelves, and the other mobile robots within a warehouse environment, is simulated using the proposed optimisation method. Finally, for comparison, the case study is simulated using the Interior-Point algorithm and Active-Set algorithm to evaluate the performances of the proposed Dual Forward–Backward Algorithm.

The results of the simulations show that, for all the implemented algorithms, the constraints given by obstacles and actuators limits are satisfied; furthermore, the trajectory tracking error is bounded and approximately constant for all the simulated mobile robots except when the constraints are active, since the robots need to deviate from the desired path to avoid obstacles or satisfy the limits of the actuators. Although the Dual Forward–Backward Algorithm has a longer computational time than the Interior-Point and Active-Set algorithms, it can solve the Model Predictive Control problem using only matrix multiplications and operations that can be easily implemented in common micro-controllers: this is advantageous since it reduces hardware complexity, development costs, and implementation time. In addition, it is possible to increase the computational efficiency

by exploiting the sparsity of the matrices. Finally, by setting the maximum number of iterations of the Dual Forward–Backward Algorithm, it is possible to define the maximum computation time required to satisfy real-time applications.

Future work will focus on implementing and testing the proposed algorithm on real mobile robots in a semi-structured environment. Further developments will exploit predictions of the future positions of each mobile robot, computed by the Model Predictive Control Algorithm, to estimate how the constraints will change in the prediction horizon. This should have the potential to significantly enhance the capability of each robot to avoid collisions, and at the same time optimise the path through earlier planning of course corrections.

**Author Contributions:** Conceptualization, D.L. and P.G.; methodology, D.L. and P.G.; software, D.L. and A.P.; validation, D.L., A.P. and L.D.M.C.D.V.; writing—original draft preparation, A.P. and D.L.; writing—review and editing, L.D.M.C.D.V., C.C. and D.G.C.; supervision, C.C.; project administration, D.G.C. All authors have read and agreed to the published version of the manuscript.

**Funding:** This research received no external funding.

**Institutional Review Board Statement:** Not applicable.

**Informed Consent Statement:** Not applicable.

**Data Availability Statement:** Not applicable.

**Acknowledgments:** The authors would like to thank Saverio Salzo and Silvia Villa for sharing their knowledge and supporting the team in the mathematical formulation of the problem.

**Conflicts of Interest:** The authors declare no conflict of interest.

## Appendix A. Equality Constraints Details

The matrices introduced to describe the dynamic equations are the following:

$$
A = \begin{bmatrix} \mathbf{I}_2 & T_s \mathbf{I}_2 \\ \mathbf{0}_{2,2} & \mathbf{I}_2 \end{bmatrix} \text{ and } B = \begin{bmatrix} T_s^2 \mathbf{I}_2 \\ T_s \mathbf{I}_2 \end{bmatrix}.
$$

The vector $b \in \mathbb{R}^{4(n_p+1)}$ and the matrix $\Phi \in \mathbb{R}^{4(n_p+1) \times n_v}$ introduced to compute the equality constraints are defined as

$$
b = \begin{bmatrix} -z_0 \\ \mathbf{0}_{4n_p,1} \end{bmatrix} \text{ and } \Phi = \begin{bmatrix} \Phi_{11} & \mathbf{0}_{4,2n_p} \\ \Phi_{21} & \Phi_{22} \end{bmatrix},
$$

where $\Phi_{11} \in \mathbb{R}^{4 \times 4(n_p+1)}$ is defined as

$$
\Phi_{11} = \begin{bmatrix} \mathbf{I}_4 & \mathbf{0}_{4,4n_p} \end{bmatrix},
$$

$\Phi_{21} \in \mathbb{R}^{4n_p \times 4(n_p+1)}$ is defined as

$$
\Phi_{21} = \begin{bmatrix} -A & \mathbf{I}_4 & \mathbf{0}_{4,4} & \cdots & & \mathbf{0}_{4,4} \\ \mathbf{0}_{4,4} & -A & \mathbf{I}_4 & \mathbf{0}_{4,4} & \cdots & \mathbf{0}_{4,4} \\ \vdots & \ddots & \ddots & \ddots & \ddots & \vdots \\ \mathbf{0}_{4,4} & \cdots & \mathbf{0}_{4,4} & -A & \mathbf{I}_4 & \mathbf{0}_{4,4} \\ \mathbf{0}_{4,4} & & \cdots & & \mathbf{0}_{4,4} & -A & \mathbf{I}_4 \end{bmatrix},
$$

and $\Phi_{22} \in \mathbb{R}^{4n_p \times 2n_p}$ is defined as

$$\Phi_{22} = \begin{bmatrix} -B & \mathbf{0}_{4,2} & \ldots & \mathbf{0}_{4,2} \\ \mathbf{0}_{4,2} & -B & \ddots & \vdots \\ \vdots & \ddots & \ddots & \mathbf{0}_{4,2} \\ \mathbf{0}_{4,2} & \ldots & \mathbf{0}_{4,2} & -B \end{bmatrix}.$$

**Appendix B. Inequality Constraints Definition**

The vector $d \in \mathbb{R}^{n_d}$ introduced to compute the inequality constraints is the following:

$$d = \begin{bmatrix} d_1^T & \ldots & d_1^T & d_2^T & d_3^T \end{bmatrix}^T,$$

where

$$d_1 = \begin{bmatrix} \langle n_1, P_1 \rangle & \ldots & \langle n_{n_a}, P_{n_a} \rangle \end{bmatrix}^T$$

is repeated $n_p + 1$ times, while

$$d_2 = \begin{bmatrix} -v_{max} \, \mathbb{1}_{2(n_p+1),1} \\ -v_{min} \, \mathbb{1}_{2(n_p+1),1} \end{bmatrix} \text{ and } d_3 = \begin{bmatrix} -u_{max} \, \mathbb{1}_{2n_p,1} \\ -u_{min} \, \mathbb{1}_{2n_p,1} \end{bmatrix}.$$

The matrix $C \in \mathbb{R}^{n_d \times n_v}$ introduced to compute the inequality constraints is described as follows:

$$C = \begin{bmatrix} C_{11} & \mathbf{0}_{n_a(n_p+1),2n_p} \\ C_{21} & \mathbf{0}_{2(n_p+1),2n_p} \\ C_{31} & \mathbf{0}_{2(n_p+1),2n_p} \\ \mathbf{0}_{4n_p,4(n_p+1)} & C_{3,2} \end{bmatrix},$$

where $C_{11} \in \mathbb{R}^{n_a(n_p+1) \times 4(n_p+1)}$ is defined as

$$C_{11} = \begin{bmatrix} V & \mathbf{0}_{n_a,4} & \ldots & \mathbf{0}_{n_a,4} \\ \mathbf{0}_{n_a,4} & V & \ddots & \vdots \\ \vdots & \ddots & \ddots & \mathbf{0}_{n_a,4} \\ \mathbf{0}_{n_a,4} & \ldots & \mathbf{0}_{n_a,4} & V \end{bmatrix},$$

with

$$V = \begin{bmatrix} -n_1^T & \mathbf{0}_{1,2} \\ \vdots & \vdots \\ -n_{n_a}^T & \mathbf{0}_{1,2} \end{bmatrix},$$

$C_{21} \in \mathbb{R}^{4(n_p+1) \times 4(n_p+1)}$ is defined as

$$C_{21} = \begin{bmatrix} M & \mathbf{0}_{2,4} & \ldots & \mathbf{0}_{2,4} \\ \mathbf{0}_{2,4} & M & \ddots & \vdots \\ \vdots & \ddots & \ddots & \mathbf{0}_{2,4} \\ \mathbf{0}_{2,4} & \ldots & \mathbf{0}_{2,4} & M \end{bmatrix},$$

with

$$M = \begin{bmatrix} \mathbf{0}_{2,2} & \mathbf{I}_2 \end{bmatrix},$$

$C_{31} \in \mathbb{R}^{4(n_p+1) \times 4(n_p+1)}$ is defined as

$$C_{31} = -C_{21},$$

and

$$C_{3,2} = \begin{bmatrix} \mathbf{I}_{2n_p} \\ -\mathbf{I}_{2n_p} \end{bmatrix}.$$

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
