# Peer review of "A Dual Forward–Backward Algorithm to Solve Convex Model Predictive Control for Obstacle Avoidance in a Logistics Scenario"

_electronics, doi:10.3390/electronics12030622_

Round 1

Reviewer 1 Report

This paper proposes a new algorithm for the warehouse robot path-finding problem (formulated into a convex problem). It is well-written, and the background is clearly presented. Good work!

From a practitioner's point of view, I'd like the authors to clarify the potential slowness brought by the DFBA algorithm. More specifically, the authors claim "the DFBA requires a relatively high number of iterations." Would it be possible that the longer convergence time makes the algorithm unusable in practice? 

Reviewer 2 Report

   This research proposes a dual forward-backward algorithm to solve convex model predictive control for obstacle avoidance in a logistics scenario. A very good effort has been made to resolve the identified problem. Paper presentation also very good. However, the following suggestions are recommended:

       Add a literature review section after introduction section and compare your finding with already published research.

       Author said dynamic model to entire mathematical description so, what are the practical implementations of the model in real time?

       Results and Discussion; the author should compare the finding of the present study with the previous study and justify for more clarity.

       Would you explicitly specify the novelty of your work? What progress against the most recent state-of-the-art similar studies was made?

       Conclusions should be amended to incorporate a broader discussion of the significance and potential application of this specific study.

       English throughout the manuscript needs to be improved.

       Article is acceptable after minor changes
